# A Novel Synthetic Antibody Library with Complementarity-Determining Region Diversities Designed for an Improved Amplification Profile

**DOI:** 10.3390/ijms23116255

**Published:** 2022-06-02

**Authors:** Xuelian Bai, Moonseon Jang, Nam Ju Lee, Thi Thu Ha Nguyen, Mooyoung Jung, Jeong Yeon Hwang, Hyunbo Shim

**Affiliations:** 1Department of Life Sciences, Ewha Womans University, Seoul 03760, Korea; back910106@naver.com (X.B.); pusrjf@naver.com (M.J.); hanguyen15490@gmail.com (T.T.H.N.); mango02@naver.com (J.Y.H.); 2Department of Bioinspired Science, Ewha Womans University, Seoul 03760, Korea; milqyou@naver.com (N.J.L.); andud8084@nate.com (M.J.)

**Keywords:** phage display, antibody library, amplification efficiency, machine learning, complementarity-determining region

## Abstract

Antibody discovery by phage display consists of two phases, i.e., the binding phase and the amplification phase. Ideally, the selection process is dominated by the former, and all the retrieved clones are amplified equally during the latter. In reality, the amplification efficiency of antibody fragments varies widely among different sequences and, after a few rounds of phage display panning, the output repertoire often includes rapidly amplified sequences with low or no binding activity, significantly diminishing the efficiency of antibody isolation. In this work, a novel synthetic single-chain variable fragment (scFv) library with complementarity-determining region (CDR) diversities aimed at improved amplification efficiency was designed and constructed. A previously reported synthetic scFv library with low, non-combinatorial CDR diversities was panned against protein A superantigen, and the library repertoires before and after the panning were analyzed by next generation sequencing. The enrichment or depletion patterns of CDR sequences after panning served as the basis for the design of the new library. Especially for CDR-H3 with a higher and more random diversity, a machine learning method was applied to predict potential fast-amplified sequences among a simulated sequence repertoire. In a direct comparison with the previous generation library, the new library performed better against a panel of antigens in terms of the number of binders isolated, the number of unique sequences, and/or the speed of binder enrichment. Our results suggest that the amplification-centric design of sequence diversity is a valid strategy for the construction of highly functional phage display antibody libraries.

## 1. Introduction

Antibody phage display is utilized widely for the development of monoclonal antibodies, due in large part to the in vitro nature of the technology, which enables the isolation of human antibodies with exquisite antigen specificities [1]. Many therapeutic antibodies have been discovered or optimized by phage display technology [2], validating its utility as a technological platform for biopharmaceutical drug development. Antibody libraries with high diversity and sophisticated designs have been constructed and shown to produce high-quality antibodies with desired functional activities suitable for therapeutic applications [3,4,5,6,7,8,9,10].

Despite its apparent technological advantages, however, antibody phage display has not entirely displaced antibody discovery from the immunized animal; on the contrary, animal-derived monoclonal antibodies (including chimeric, humanized, and fully human antibodies) constitute a majority of therapeutic antibodies used currently in clinic or under clinical evaluation [11]. It has been suggested that the eukaryotic quality control machineries may favor antibody clones with better physicochemical properties and expression efficiency [12], critical attributes for the successful development of therapeutic antibodies. Another factor that distinguishes the in vivo antibody generation process from the in vitro phage display approach is the linkage between binding and amplification. B cell activation and proliferation require antigen binding to a functional B cell receptor (BCR) and the subsequent signaling events, and the absence of in-frame BCR or the lack of antigen binding to BCR results in B cell apoptosis [13]. In theory, a similar mechanism is applied to antibody phage display; antigen-binding phage clones are selected and amplified in *Escherichia coli* host cells, whereas non-binders are washed out and discarded. However, unlike the B cell activation by BCR signaling, the binding and amplification steps of phage display are separated spatially and temporally. As a result, a large number of nonbinding clones are retrieved and amplified, and, because some of them—especially non-producing clones with early stop codons—have a sufficient growth advantage to compensate for the loss in numbers during wash cycles, they may overwhelm the output after multiple rounds of biopanning.

Rapid amplification of nonbinding, fast-growing clones can be minimized by a number of strategies, most notably the engineered helper phages lacking a functional copy of *gIII* [14,15,16,17,18]. Since infection of *Escherichia coli* by M13 bacteriophage requires the minor coat protein pIII, phages rescued by such an engineered helper phage cannot infect the host unless it displays a foreign protein fused to a functional, phagemid-derived pIII. Although this is an effective strategy for excluding non-producing clones, it cannot prevent the amplification of highly productive nonbinders; furthermore, such characteristics as the multivalent display, low production titer, or incompatibility with the common phage display features (e.g., the use of a phagemid with truncated *gIII* or an amber stop codon) may limit wider applicability of this strategy [15]. Another, probably more fundamental, solution to the problem is to design and construct a library consisting of antibody clones with desirable physicochemical profiles. For example, hundreds of human VH/VL pairs were screened for stability, expression, display level, and monomeric contents in order to design ideal frameworks for a synthetic human Fab library [9].

In this report, we attempted to alleviate the problem of amplification bias by designing an antibody library consisting of clones with a better amplification profile and that are thus less likely to be outcompeted by non-binders during the amplification phase of biopanning (Figure 1). Using a previously constructed library with low, non-combinatorial complementarity-determining region (CDR) diversities (OPAL-S) [3], single-chain variable fragment (scFv) clones that were amplified and/or displayed more efficiently were enriched by panning against protein A from *Staphylocucccus aureus*. Protein A is primarily known for its binding activity to the Fc portion of immunoglobulins [19], but it also binds to variable domains belonging to human VH3 subgroup [20]; this unique characteristic has been utilized to select well-folded and highly displayed antibody fragments from phage libraries [10,21,22]. The effect of CDR sequences on the panning enrichment of antibody clones was evaluated by comparing the next generation sequencing (NGS) repertoires of the library before and after panning. CDR sequences for the new library were designed based on the NGS analysis and, especially for CDR-H3, machine learning (ML) models were trained from the sequencing data and utilized to predict fast-amplified sequences. The new library, OPAL-T, generally performed better than OPAL-S when tested against a panel of antigens, highlighting the effectiveness of the antibody library design strategy focused on amplification efficiency.

## 2. Results

### 2.1. NGS Analysis of Pre- and Post-Panning Phage Antibody Libraries

Previously we reported the design and construction of a synthetic scFv library (OPAL-S) with predefined CDR diversities [3]. The library is based on the VH3-23 (DP47) heavy-chain scaffold, which binds to protein A from *Staphylococcus aureus*; therefore, panning of the library against protein A would enrich clones that are amplified rapidly and/or displayed efficiently. Since each CDR of OPAL-S consists of a relatively small number of unique, predefined sequences (~8000 for CDR-H3 and ~1500 for the other CDRs), the change in the relative abundance of virtually all individual CDR sequences can be determined by NGS analysis. The sequence characteristics of the panning-enriched CDR sequences may provide useful information for the design of a novel synthetic antibody library with a more efficient amplification profile.

After three rounds of panning against protein A, phagemid DNA was isolated from the panning outputs and the variable heavy (VH) and light (VL and VK) domains were amplified by PCR. Paired-end sequencing of the amplified DNA by Illumina MiSeq™ yielded 1.5–2.3 million sequence reads per domain, from which CDR sequences were extracted. The non-H3 CDRs (CDR-H1, H2, L1, L2, and L3) of OPAL-S were designed by simulating somatic hypermutations on human germline CDR sequences, allowing the identification of the germline origins of the individual NGS-analyzed CDR sequences. When the relative abundances of each ancestral germline CDR sequence in pre- and post-panning repertoires were compared, sequences from several V gene families were enriched or depleted, twofold or more, most conspicuously in CDR-H2 (Figure 2). In CDR-H2, the VH3 germline family sequences were clearly enriched after panning, whereas the VH1, VH4, and VH5 families were depleted (Figure 2b). This result suggests that the VH3-23 scaffold is more compatible with CDR-H2 originating from its own family (VH3) and that scFv clones with CDR-H2 resembling that of the VH3 family are amplified preferentially during panning.

### 2.2. CDR-H3 Sequence Analysis

The panning enrichment pattern of CDR-H3 was not analyzed according to germline origin, as was done for the other CDRs, because the CDR-H3 diversity of OPAL-S was designed not by simulating the immune system’s VDJ recombination and somatic hypermutations, but by mimicking the amino acid frequencies at each position of natural CDR-H3. Instead, the panning enrichment of each individually designed CDR-H3 sequence (of which there are <8000 in OPAL-S) was tracked by NGS, which typically produces millions of paired-end reads. Only approximately half of the NGS-analyzed CDR-H3 sequences matched the original design of OPAL-S, due to errors introduced during parallel synthesis [3]; the non-designed sequences were excluded from the subsequent analyses because they were diverse in number but low in abundance, and generally provided relatively little information about the panning enrichment pattern of CDR-H3. The change in the relative frequency of CDR-H3 sequences before and after the panning varied widely, and increased or decreased by 20-fold or greater (log[fold enrichment] >1.3 or <−1.3) for some sequences (Figure 3a). Since OPAL-S is a single-scaffold scFv library and other CDRs do not seem to influence the panning enrichment behavior of the library significantly (except CDR-H2; see above), the result suggests that CDR-H3 sequences may be important determinants of the amplification of phage-displayed scFv clones.

### 2.3. Optimization of CDR-H3 Sequences by ML

Closer inspection of the CDR-H3 sequences and their panning enrichment patterns revealed that highly enriched sequences were rich in small, polar, or charged residues, whereas highly depleted clones were rich in bulky hydrophobic residues (Figure 3b). The amino acid usage pattern of highly enriched clones more closely resembles that of natural human CDR-H3 sequences upon which OPAL-S was designed, although deviations from the natural repertoire (e.g., position 98) were also observed. However, extracting rules for designing optimized CDR-H3 sequences from the NGS analysis results was not a straightforward process because this general trend was rather weak and position-dependent (Figure 3c), and not reliably predictive of the panning enrichment outcome of individual sequences. Therefore, rather than attempting to rationally design optimized CDR-H3 sequences, randomly generated CDR-H3 sequences were subjected to virtual screening to identify potential “fast-amplifiers,” using ML models trained on the NGS data. For each CDR-H3 length, a different ML model was trained. The post-panning enrichment scores (ESs) of individual CDR-H3 sequences were calculated from the NGS data, and 70% of the sequences and their ESs were used as a training set. The validation of the trained models using the remaining 30% of the data showed that, on average, the predicted ESs were consistently proportional to the actual ESs, although the prediction accuracy varied widely among sequences (Figure 4a–h). 

Thousands of CDR-H3 sequences for each length ranging from 9 to 16 amino acids were simulated as described previously [3], based on the amino acid usage of natural human antibodies and excluding deleterious post-translational modification motifs. These sequences were evaluated by the abovementioned ML models, and those with the predicted ES > 0 (i.e., sequences predicted to be enriched by panning) were chosen for the construction of the scFv library.

**Figure 3 ijms-23-06255-f003:**
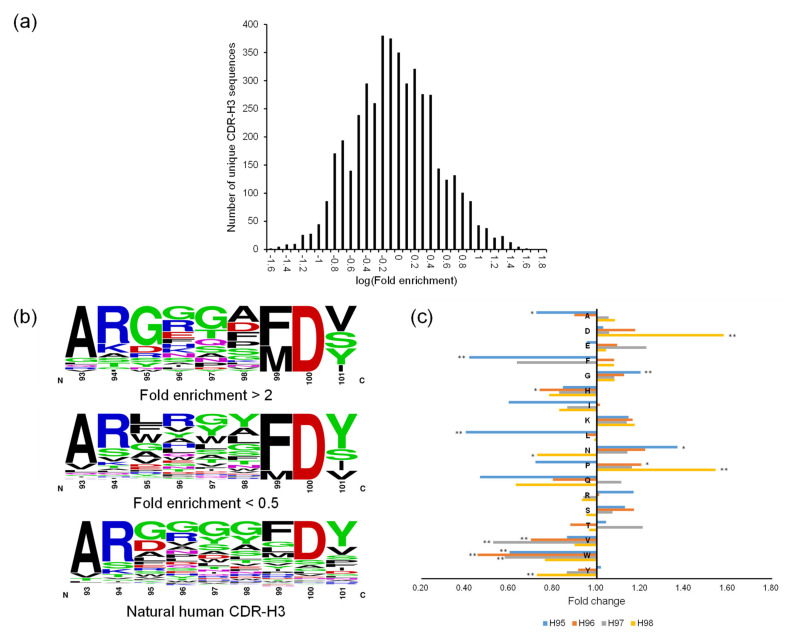
(**a**) CDR-H3 sequence read counts within the range of enrichment ratios (in logarithmic increment of 0.1) are presented as a histogram. A representative plot for CDR-H3 length 12 is shown here. (**b**) Logograms of CDR-H3 (length 9 aa) repertoires enriched (**top**) or depleted (**bottom**) more than twofold after the panning against protein A. Sequence logograms were created by WebLogo [21]. (**c**) Change in abundance of each of 18 aa (except Cys and Met, which had been excluded in the library CDR design) at each position from H95 to H98 of CDR-H3 (length 9 aa) before and after the panning against protein A. In (**b**) and (**c**), CDR positions are numbered according to Kabat numbering scheme. *: *p* < 0.05, **: *p* < 0.01.

### 2.4. In silico Deimmunization

NetMHCIIpan-3.1 was used to identify potential T cell epitopes among the designed CDR sequences that could cause immunogenicity of therapeutic antibodies [22]. The flanking 8-mer amino acid sequences from the adjoining framework regions were added to both ends of the individual CDR sequences, and overlapping 9-mer fragments of these sequences were evaluated for possible binding to 20 of the most frequent HLA-DRB alleles in Caucasian, African, Asian, and Hispanic populations [23]. CDR sequences that were predicted to be strong binders (within the top 0.5% predicted affinity among 20,000 random natural human peptides) were excluded from the final library design.

**Figure 4 ijms-23-06255-f004:**
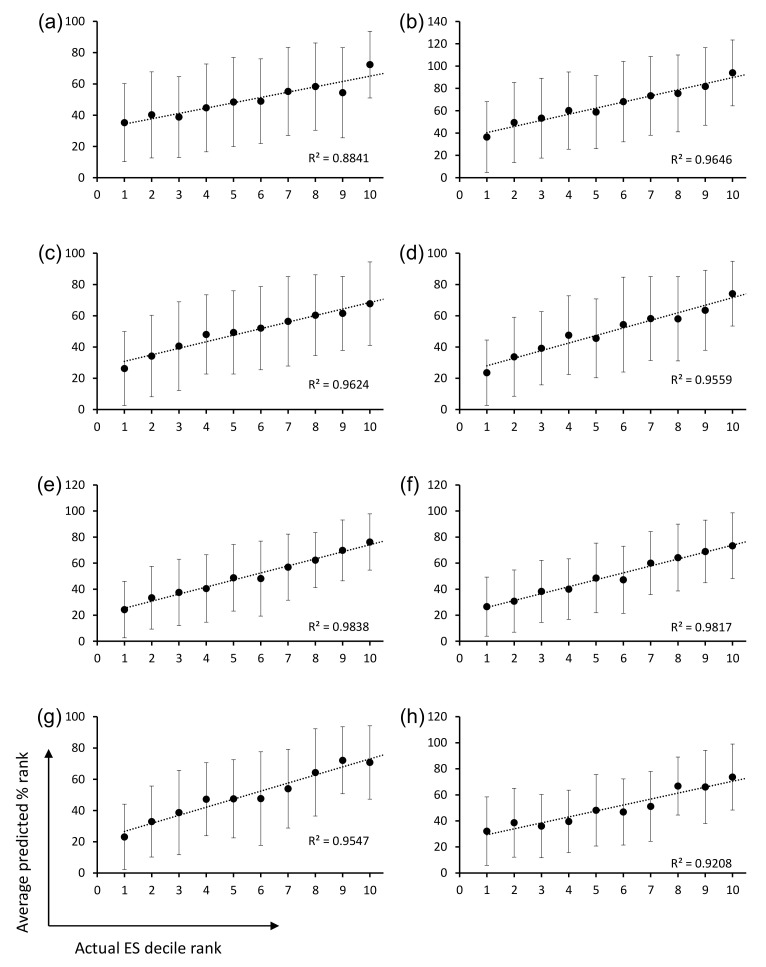
Plots of the average predicted enrichment score (ES) percentile rank of CDR-H3 sequences belonging to each decile rank of actual ES. The results for CDR-H3 sequences of length (**a**) 9 aa, (**b**) 10 aa, (**c**) 11 aa, (**d**) 12 aa, (**e**) 13 aa, (**f**) 14 aa, (**g**) 15 aa, and (**h**) 16 aa are shown. Error bars indicate standard deviations of the predicted ES.

### 2.5. Construction of OPAL-T

The finalized CDR oligonucleotide sequences were array-synthesized (27,911 sequences in total). The oligonucleotides were 136 nucleotides in length, including adjoining framework sequences and common adaptor sequences for PCR amplification at both ends. The synthesized oligonucleotide pool was amplified by PCR using a pair of common primers specific for the adaptor sequences, and each CDR (CDR-H1, H2, H3, L1, L2, and L3) was re-amplified from the amplified pool using framework-specific primers. CDR-H3 was designed with a wide range of lengths (9 to 16 amino acids [aa]). In the previous generation library, OPAL-S, we observed that shorter CDR-H3s were overrepresented in the final library [3], likely due to more synthetic errors in longer CDRs and the growth advantage for the clones with shorter CDR-H3s. In an effort to reduce this bias, the amplified CDR-H3 oligonucleotides were separated by length using polyacrylamide gel electrophoresis (PAGE), and CDR-H3 oligonucleotides with different lengths were retrieved separately from excised gel bands.

Parallel DNA synthesis introduces errors, among which nucleotide insertions and deletions (indels) are most prominent [24]. Indels introduce frameshifts that lead to premature termination of translation and reduce the effective diversity of antibody libraries. In order to minimize the occurrence of indels in the final library, single-CDR libraries (i.e., focused libraries in which only a single CDR is diversified using the amplified oligo pool) were constructed and subjected to a round of panning against protein A. Since protein A binds to the VH3-23 scaffold, clones that express and display full-length scFv are enriched preferentially, and CDR sequences with indels are depleted concomitantly. The selected CDR sequences from the post-panning single-CDR libraries were assembled to construct the final scFv library, OPAL-T. Sixteen sub-libraries were assembled separately, transformed, and stored according to their light chain class (kappa or lambda) and the length of CDR-H3 (9 to 16 aa). The combined library sizes for OPAL-T-κ and OPAL-T-λ were 3.4 × 10^9^ and 1.1 × 10^10^, respectively (Table 1).

### 2.6. Validation of OPAL-T

Before proofread panning of single-CDR libraries on protein A, the percentages of scFv-expressing clones were 23–86%, which increased to 82–100% after panning, as determined by dot blot assay (Figure 5a,b). Assuming that 90% of sequences are in-frame for each CDR after proofread panning, it is expected that the final library with six such CDRs has ~50% dot blot-positive ratio (0.9^6^ = 0.53). Indeed, when these proofread CDRs were combined into the final OPAL-T library, the overall percentage of the clones expressing soluble scFv was estimated to be 40–50% (Table 1). Although this number is lower than some of the reported natural or synthetic antibody libraries due to the low-fidelity nature of parallel oligonucleotide synthesis [7,9,25], the estimated size of the final library with 1.4 × 10^10^ transformants supported testing for the generation of high-quality, target-specific binding agents.

In order to test whether the design strategy applied to the construction of OPAL-T (i.e., NGS analysis of OPAL-S and ML based in silico screening for rapidly amplified CDR sequences) resulted in a better-performing antibody library, a side-by-side comparison of the two libraries was conducted using several antigens (Table 2). OPAL-T performed equivalently to or better than OPAL-S for most of the antigens tested, suggesting that the improved design of OPAL-T allowed more efficient enrichment of target-binding clones during biopanning. Panning campaigns against antigens such as cysteinyl tRNA synthetase (CARS) were highly efficient and yielded a high percentage of binders from both libraries after three rounds of panning, whereas for some other antigens (e.g., hNinj1, BCMA, or CD22), OPAL-T produced higher percentages of binders than OPAL-S. As expected, binder sequences identified in round 3 were also typically found in round 4, but the output clones could be highly diverse; for example, none of the eight unique binders isolated from round 4 panning of OPAL-T-λ against CARS was identical to any of seven unique binders from round 3. Of 74 binders sequenced from the round 4 pannings of OPAL-T, 46 had unique sequences (62%), compared to 34/93 unique sequences (37%) for OPAL-S.

In total, 14 target-binding scFvs against 6 different antigens were chosen randomly and purified, and their affinities were analyzed by surface plasmon resonance (SPR) measurement. Dissociation constants (*K*_D_) ranging from 20 to 360 nM were obtained, with a mean *K*_D_ = 84 nM (Table 3, Figure 6 and Appendix A). The monovalent affinities of scFvs isolated from OPAL-T are comparable to the affinities of antibodies produced from other sources [5,26].

Unique binders from the panning against hNinj1 were expressed from *E. coli*, purified by immobilized metal ion affinity chromatography, and analyzed by size exclusion chromatography (SEC). The expression levels and the monomeric percentages of clones from OPAL-T seem to be higher than those from OPAL-S (Figure 7 and Table 4), although larger sample sizes are necessary to draw a definitive conclusion. Since the two libraries are identical in their framework amino acid sequences (VH3-23, Vκ3-20, and Vλ1-47), it is possible that the design of OPAL-T CDRs optimized for better amplification also has some indirect effects on the expression and monomeric solubility of scFv.

## 3. Discussion

During panning of a phage-displayed antibody library, the selection outcome is determined not only by the binding affinity to the target antigen, but also in large part by the display level and/or growth rates of the clones. The diversity of the panning output may be reduced greatly as a result, because of the preferential enrichment of a small number of binders with significant display/growth advantage, or the faster amplification of some non-binders or non-producers. Even within a synthetic antibody library that employs a fixed framework scaffold, wide variation in the amplification efficiency among the clones was observed, suggesting that CDR sequences have a significant influence on the enrichment of antibody clones during biopanning.

A synthetic scFv library with pre-defined CDR sequences was designed and constructed previously (OPAL-S) [3]. It should be noted that although the reported size of OPAL-S was 8 × 10^8^, the library used in this study was expanded to 1.3 × 10^10^ through additional transformations and is comparable in size to OPAL-T (1.4 × 10^10^_,_ Table 1). Since the designed diversity of each CDR in this library was between 10^3^–10^4^, we were able to track the change in relative abundance of every predefined CDR sequence before and after biopanning against protein A by NGS. Non-H3 CDR sequences of OPAL-S were designed individually by simulating somatic hypermutations to the human germline CDR sequences, and thus they were traceable to their germline ancestors. From the analysis of panning enrichment patterns of the CDR sequences, with the exception of CDR-H2, the germline origin of CDR sequences appeared not to have marked influence on the amplification of scFv clones in the context of the VH3-23, Vκ3-20, and Vλ1-47 scaffolds. This conclusion does not necessarily mean that a specific CDR sequence has no influence on the amplification of the scFv; rather, it suggests that no major intrinsic incompatibility exists between those scaffold sequences and the CDRs from different germline genes or families, in terms of the levels of phage display and amplification.

The ML models were trained and validated for CDR-H3 using the OPAL-S panning enrichment NGS data set. The trained ML models predicted the average enrichment trend of CDR-H3 sequences across the full decile range of actual ESs, although the wide variation in the prediction accuracy for individual sequences resulted in large error bars and shallow slopes (Figure 4). The moderate accuracy of the prediction is probably due in part to the fact that the CDR-H3 sequences are unlikely to be the sole determinant of the panning amplification outcome, and also in part to the choice and the performance of ML algorithm and the quality of real-world data upon which ML models were trained and validated [26].

In several recent studies, ML has been applied to antibody design and optimization. For example, CDR-H3 sequences with high affinity and specificity were predicted by a ML method trained on panning enrichment data against a target antigen [27]. The high prediction accuracy of this study may be attributed in part to the fact that only CDR-H3 was diversified, thus eliminating other uncertainties within the library, and that the ML model was trained to predict a well-defined, quantifiable property (i.e., target binding affinity). In another example, ML prediction of hot spot CDR residues led to the design, construction, and validation of scFv libraries that generated antibodies to diverse epitopes on a protein antigen with affinities comparable to those of in vivo affinity-matured antibodies [28]. The approach reported in our study utilized data obtained from a library that was panned against protein A and thus selected for amplification efficiency. While the binding specificity and affinity, which abovementioned ML-based library designs aimed to improve, depend primarily on complementary interactions between the epitope and the paratope, amplification efficiency depends on multiple factors, such as display level, protein folding, stability, and the host cell interaction of scFv clones. The complex nature of panning amplification may have contributed to the moderate prediction accuracy of the ML models. Nonetheless, the new OPAL-T library, designed and constructed based on the prediction, yielded target binders more efficiently than OPAL-S, even though the two libraries were constructed on identical frameworks and similar in size (~10^10^; see above) and in the design of CDR diversity (predefined, non-combinatorial diversity). It is likely that the NGS- and ML-based optimization of the CDR sequences contributed distinctly to the improved performance of OPAL-T over OPAL-S.

The design goal of OPAL-T was to construct an antibody library with optimized amplification profile; thus, the binders were enriched faster in OPAL-T than in OPAL-S, but once individual binders were enriched by panning and isolated from the libraries, the affinities of OPAL-T clones were not significantly better than those of OPAL-S clones. For example, the average affinity (*K*_D_) of 14 OPAL-T clones against five different antigens was 84 nM, compared with the average *K*_D_ of 29 nM for 10 OPAL-S clones against three different antigens [3]; the average affinity of binders to SARS, for which both libraries were evaluated by SPR, was 24 nM and 34 nM for OPAL-T and OPAL-S, respectively. On the other hand, the expression and monomeric ratio of the binders isolated from OPAL-T seemed to be higher and more uniform than those from OPAL-S. Apart from different codon optimization schemes employed (see Section 4.6) which might influence expression but not monomeric percentage, the two libraries have identical framework sequences. Since the efficient amplification of phage antibody clones requires fast growth, good expression, and/or proper folding, it is plausible that the amplification-centric design of CDR diversity also has contributed to the improved expression and aggregation resistance of OPAL-T-derived clones.

The diversity of OPAL-T was generated by massive parallel synthesis of oligonucleotides. The synthetic fidelity of nucleotide addition is very high (up to 99.5%), but for longer oligonucleotides, the full-length purity (i.e., the proportion of oligonucleotide molecules with correct length) can be much lower [24], resulting in frameshifts and early termination of translation. In order to minimize the introduction of unintended frameshifts to the final library, single-CDR scFv libraries were prepared and subjected to a single round of panning against protein A. Although this additional selection step significantly improved the proportion of clones that were expressed solubly in the library, it also may have introduced additional bias against CDR sequences that were amplified slowly. Given the observation that the scaffold sequences used for the construction of OPAL-T appear not to have strong preference for specific CDR origins over others (except for CDR-H2 and CDR-H3, for which the diversities were designed to minimize potential biases), the biases caused by the proofread panning may be insignificant. Nonetheless, library construction will benefit by future development in oligonucleotide synthesis and purification technologies, which are expected to provide much higher synthetic fidelity.

To summarize, CDR diversities for a novel scFv library aimed at improved amplification efficiency were designed based on sequence enrichment data and ML analysis. The newly constructed OPAL-T library was validated using a panel of test antigens and demonstrated superior performance in terms of the binder enrichment efficiency compared to OPAL-S, which provided the sequencing data used for its design. Library design will be improved by more detailed analysis of the effects of various somatic hypermutations on amplification efficiency or the compatibility of diverse CDRs combined in a single scFv molecule, as well as by advances in artificial intelligence and parallel oligonucleotide synthesis technologies.

## 4. Materials and Methods

### 4.1. Materials

Oligonucleotide libraries for CDRs were synthesized by LC Sciences (Houston, TX, USA; Oligomix^TM^). Wild-type protein A from *S. aureus* was purchased from Thermo Fisher Scientific (Waltham, MA, USA; Pierce^TM^ 21181). Variable domain genes for library construction were codon-optimized and synthesized (GenScript, Piscataway, NJ, US). CARS, SARS, and AIMP1 were kindly provided by Prof. Sunghoon Kim (Yonsei University, Incheon, Korea). Other reagents, antigens, and consumables used in this study were purchased from readily available commercial sources.

### 4.2. Panning of Phage Antibody Libraries

All experiments were performed at room temperature unless stated otherwise. An immunotube (Nunc 470319, Thermo Scientific) was coated with antigen (1–10 μg/mL in PBS) for 1 h and subsequently blocked with mPBST (PBS with 3% skim milk and 0.05% Tween-20) for another hour. Phage library (10^12^ cfu) preincubated for 1 h in mPBST was added to the tube and allowed to bind for 2 h. Unbound phages were removed by washing the tube with PBST (PBS with 0.05% Tween-20) three times, and bound phages were eluted by 1 mL of 100 mM triethylamine solution. The eluted phage solution was neutralized with 0.5 mL of 1 M Tris-HCl (pH 8.0), and 8.5 mL of mid-log phase TG1 *E. coli* culture in SB medium (Super Broth; 3% tryptone, 2% yeast extract, 1% MOPS, pH adjusted to 7.0) was added and incubated for 1 h at 37 °C with slow shaking at 120 rpm. The infected bacteria were collected by centrifugation and plated on 150-mm diameter LB-agarose plate supplemented with ampicillin (100 μg/mL) and 2% glucose (*w*/*v*).

Overnight bacterial growth was recovered from the plate by adding 5 mL of SB medium to the plate and scraping the growth with a flame-sterilized glass spreader. Fifty microliters of the collected bacterial suspension was added to 20 mL of SB medium supplemented with ampicillin (100 μg/mL). After incubating for ~2 h, when the culture became visibly turbid (OD600~0.7), 10^11^ pfu of VCSM13 helper phage was added. The culture was further incubated for 1 h at 37 °C with slow shaking at 120 rpm to allow helper phage superinfection. Kanamycin (70 μg/mL) was added, and the culture was incubated overnight (~16 h) at 30 °C with shaking at 220 rpm. The culture was centrifuged and the supernatant was transferred to a fresh 50 mL conical tube. Amplified phages were precipitated by adding 5 mL of 5× PEG precipitation solution (20% [*w*/*v*] PEG 8000, 15% [*w*/*v*] NaCl) and incubating the mixture on ice for >30 min. Precipitated phages were collected by centrifugation (14,000× *g* for 15 min), resuspended in 0.3 mL PBS, and used for the next round of selection.

### 4.3. NGS Analysis of OPAL-S

Sequence analysis of the pre-selection OPAL-S was described previously [3]. For sequence analysis of the post-selection OPAL-S, phagemid DNAs from the output of the third panning round against protein A were isolated, and variable domain repertoires were amplified by PCR using a library-specific primer pair with adapter sequences for sequence analysis with Illumina MiSeq™. VH and VL sequences of the scFv library were obtained by 300 bp paired-end sequencing, and CDR sequences were extracted using an in-house Python script. Invariable framework region sequences of OPAL-S were identified in each variable domain sequence, and the sequence between two adjacent framework regions was identified as CDR. Each of the retrieved CDR sequences (except CDR-H3) was matched to the most similar human germline CDR sequence.

### 4.4. ML-Based Optimization of CDR-H3 Sequences

The enrichment score (ES) of each CDR-H3 sequence in the pre- and post-panning NGS repertoires was calculated using the formula:$$\mathrm{ES}=\;\log_{2}\frac{n_{post}/N_{post}}{n_{pre}/N_{pre}}\times \sqrt{\frac{n_{post}+n_{pre}}{median\left(n_{post}\right)+median\left(n_{pre}\right)}}$$
where *n_pre_* and *n_post_* are the read counts of the specific CDR-H3 sequence in pre- and post-panning repertoires, respectively, and *N_pre_* and *N_post_* are the total read counts of the pre- and post-panning libraries, respectively.

The calculated ESs of CDR-H3 sequences were used to train ML models using a web-based ML service [29]. Specifically, a comma-separated values (.csv) file comprising the CDR sequences, their ESs, and the amino acid residues by position was created for each CDR-H3 length to train an ML model. The CDR-H3 sequence, its individual amino acid residues, and the ES were given the attributes of text, categorical, and numeric, respectively, and the ES was set as a target. The following parameters were used: maximum ML model size, 100 MB; maximum number of data passes, 100; L2 regularization, mild (10^−6^). Seventy percent of ES data was used as the training data set, and the remaining 30% was used as the evaluation data set to validate the models. Subsequently, CDR-H3 sequences simulated as described in a previous report [3] were evaluated using the validated ML model, and sequences with predicted ES > 0 were selected for parallel oligonucleotide synthesis.

### 4.5. In Silico Deimmunization

The designed CDR sequences were then evaluated for their potential binding to human MHC class 2 molecules using netMHCIIpan-3.1 software [22]. Alleles used for evaluation were DRB1*01:01, DRB1*03:01, DRB1*03:02, DRB1*04:01, DRB1*04:04, DRB1*04:05, DRB1*07:01, DRB1*08:02, DRB1*08:03, DRB1*09:01, DRB1*11:01, DRB1*13:01, DRB1*13:02, DRB1*12:02, DRB1*14:01, DRB1*15:01, DRB1*15:03, DRB3*01:01, DRB4*01:01, and DRB5*01:01, which, when combined, represent 81.2%, 75.1, 71.3, and 61.7% of Caucasian, East Asian (Korean), Black, and Hispanic populations, respectively [23]. Overlapping 9-aa fragments of the CDR sequences (with the adjoining 8-aa framework sequences on both sides) were analyzed, and the CDR sequences predicted to be strong binders (0.5% threshold) to any of the 20 alleles were discarded.

### 4.6. Library Construction

The adjoining framework sequences and PCR adaptor sequences were added to both ends of the designed CDR sequences to produce a collection of 136-mer oligonucleotide sequences for parallel synthesis. In total, 27,911 sequences were synthesized (LC Sciences; purity unmeasurable due to the minute quantity and the mixed nature of the oligo pool). The oligonucleotides were amplified by PCR using a pair of primers specific for the adaptor sequences, and individual CDRs (CDR-H1-H3 and L1-L3 for kappa and lambda classes) were further amplified using primers specific for the framework sequences. For CDR-H3, PCR-amplified oligonucleotides were separated by length using PAGE.

Variable domain genes for the library were codon-optimized for human, and rare codons in *E. coli* were subsequently replaced with codons non-rare in both human and *E. coli* when possible. Codon-optimized VH3-23/JH4-(G4S)3-Vκ3-20/Jκ1 and VH3-23/JH4-(G4S)3-Vλ1-47/Jλ2 scaffolds were synthesized (GenScript), to which an individual CDR repertoire was inserted by overlap-extension PCR to make 16 single-CDR variant libraries (eight for CDR-H3 of different lengths, plus one each for other CDRs of heavy, kappa, and lambda variable domains). These libraries were subjected to a single round of panning against protein A to remove CDR sequences with synthetic errors that resulted in stop codons, indels, and frameshifts. The proofread CDR sequences were amplified from the panning output and assembled with framework regions by a series of overlap extension PCRs. The assembled scFv repertoires were ligated into pCOMB3X phagemid vector and transformed to electrocompetent TG1 *E. coli* (Lucigen, Middleton, WI, USA) to produce final phagemid libraries [3]. PCR schemes and primer sequences for construction of the library are provided in Appendix A, respectively.

### 4.7. Dot Blotting

Single colonies of library clones in TG1 *E. coli* were grown in 200 μL of SB-ampicillin in a 96 well plate for 4 h at 37 °C. IPTG was added to each well (1 mM final concentration) and the plate was incubated overnight at 30 °C. Next day, the plate was centrifuged (2500× *g* for 15 min at 4 °C) and supernatants were removed. Bacterial cell pellets were resuspended in 60 μL/well of ice-cold 1× TES buffer (50 mM Tris, 1 mM EDTA, 20% [*w*/*v*] sucrose, pH 8.0). Subsequently 90 μL/well of ice-cold 0.2× TES was added and mixed well, and the plate was incubated on ice for 30 min. After centrifugation (2500× *g* for 15 min at 4 °C), 1 μL of the supernatant containing periplasmic extract from each well was applied to a piece of nitrocellulose membrane. The membrane was allowed to dry, blocked with mPBST, and incubated with 1:3000 dilution of anti-HA-HRP antibody (sc7392, Santa Cruz Biotechnology, Dallas, TX, USA) in mPBST for 1 h at room temperature with slow shaking. The dot-blot membrane was detected by enhanced chemiluminescence (ECL) reagent.

### 4.8. Purification of scFv

Binder scFv clones from panning output in TG1 *E. coli* were grown in 20 mL of SB-ampicillin (100 μg/mL) at 37 °C until OD_600_ = 0.7. IPTG was added (1 mM final concentration), and the induced culture was incubated overnight at 30 °C with shaking (200 rpm). Next day, the culture was centrifuged (2500× *g* for 15 min at 4 °C), supernatant was removed, and the cell pellet was resuspended in 1 mL of ice-cold 1× TES buffer. Subsequently 1.5 mL of ice-cold 0.2× TES was added and mixed well, and the mixture was incubated in ice for 30 min. After centrifugation (14,000× *g* for 15 min at 4 °C), the supernatant was transferred to a clean 15 mL conical tube, 5 mM final concentration of MgCl_2_ was added to quench EDTA, and 100 μL of Ni-NTA agarose beads suspension (EMD Millipore) was added. The mixture was slowly rotated for 1 h to allow binding of scFv to the beads at room temperature, centrifuged briefly, and the precipitated beads were washed twice with 1 mL of wash buffer (PBS with 5 mM imidazole, pH 7.4). Finally, scFv was eluted by adding 200 μL fractions of elution buffer (PBS with 200 mM imidazole, pH 7.4).

For larger scale expression and purification, 400 mL of SB-ampicillin was inoculated with 5 mL seed culture. After initial growth and overnight induction as above, periplasmic extract was obtained using 16 mL of 1× TES and 24 mL of 0.2× TES. The extract after MgCl_2_ treatment was applied to 0.5 mL of Ni-NTA agarose beads in a gravity flow column (Poly-Prep, Bio-Rad, Hercules, CA, USA). The beads were washed twice with 10 mL of wash buffer, and eluted in 0.5 mL fractions.

### 4.9. Surface Plasmon Resonance

SPR analysis was performed by using Biacore 3000 system. The antigen in sodium acetate solution (pH 4.0~5.5) was immobilized (target RU: 800~1200) on a CM5 chip (Cytiva) at a flow rate of 5 μL/min following the standard amine coupling protocol provided by the manufacturer, and purified scFv antibodies at 5–6 different concentrations were injected at a flow rate of 40 μL/min. The data was fitted to 1:1 Langmuir binding model or 1:1 binding with drifting baseline model using BiaEvaluation software to obtain kinetic parameters and dissociation constants.

### 4.10. Size Exclusion Chromatography

Size exclusion chromatography analysis was performed on ÄKTA Pure system (Cytiva, Marlborough, MA, USA) using Superdex 200 Increase 10/300 GL column (Cytiva). Purfied scFv (500 μL of 1–2 mg/mL concentration) was injected and run on degassed PBS buffer at a flow rate of 0.75 mL/min, and the eluted protein was monitored by absorbance at 280 nm.

## Figures and Tables

**Figure 1 ijms-23-06255-f001:**
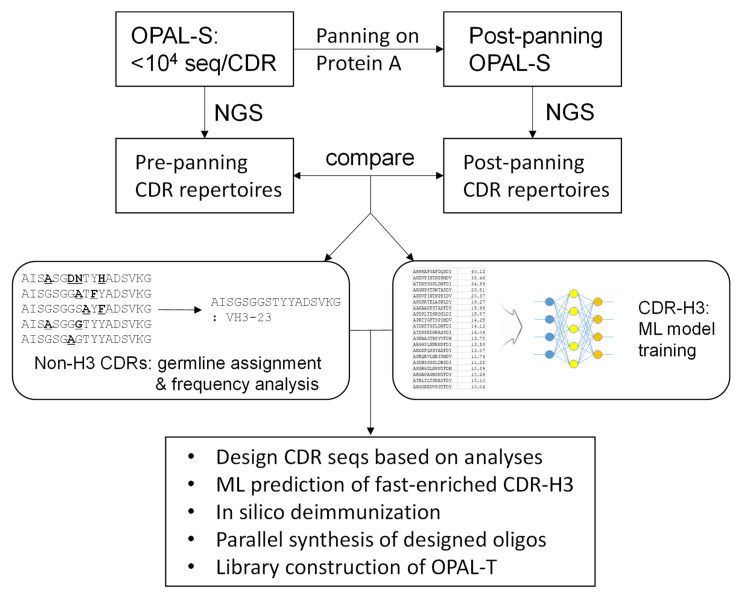
Library design process for OPAL-T. Non-combinatorial CDR repertoires of the previous generation scFv library OPAL-S before and after panning against protein A were analyzed by NGS. The enrichment pattern of CDR sequences was applied to the design of OPAL-T. For CDR-H3 especially, ML models were trained on the NGS data and utilized for the design of improved CDR-H3 diversity.

**Figure 2 ijms-23-06255-f002:**
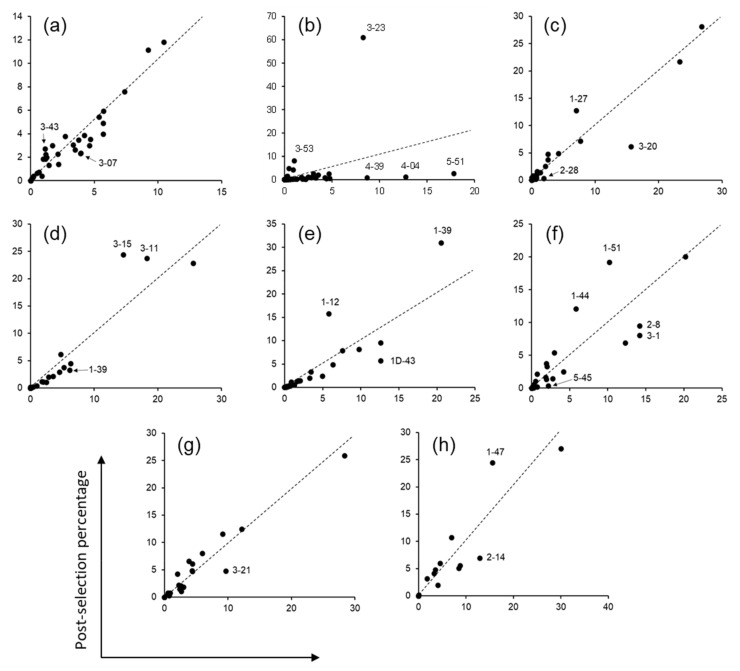
Panning enrichment of CDR sequences from various human variable gene germline origins. Data points indicate the frequencies of CDR sequences that share a germline ancestor in the pre- and post-panning NGS repertoires. (**a**) CDR-H1, (**b**) CDR-H2, (**c**) CDR-κL1, (**d**) CDR-κL2, (**e**) CDR-κL3, (**f**) CDR-λL1, (**g**) CDR-λL2, and (**h**) CDR-λL3. Germlines enriched or depleted approximately twofold or more are indicated. Dashed lines indicate no change in frequency after panning (slope = 1).

**Figure 5 ijms-23-06255-f005:**
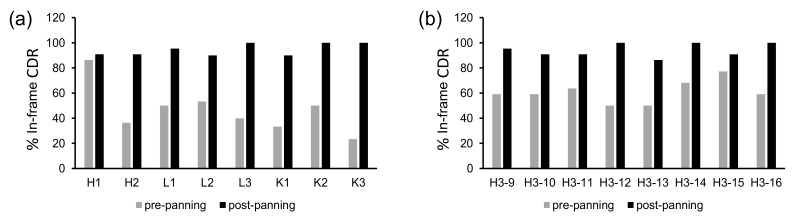
Percent proportions of in-frame CDRs before and after single-round panning against protein A of single-CDR scFv libraries, determined by dot blot assay. (**a**) Non-H3 CDRs, (**b**) CDR-H3s of different lengths from 9 to 16 aa (Kabat definition).

**Figure 6 ijms-23-06255-f006:**
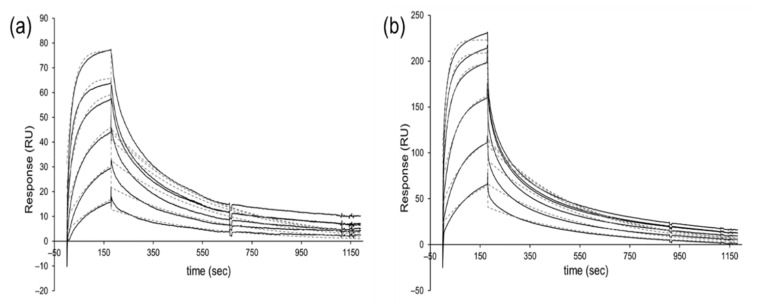
Representative SPR sensorgrams (solid lines) and 1:1 Langmuir binding fitting (dashed lines) of (**a**) AIMP1-D4 (*K*_D_ = 76 nM; [scFv] = 67–1070 nM) and (**b**) BCMA-B5 (*K*_D_ = 37 nM; [scFv] = 40–640 nM).

**Figure 7 ijms-23-06255-f007:**
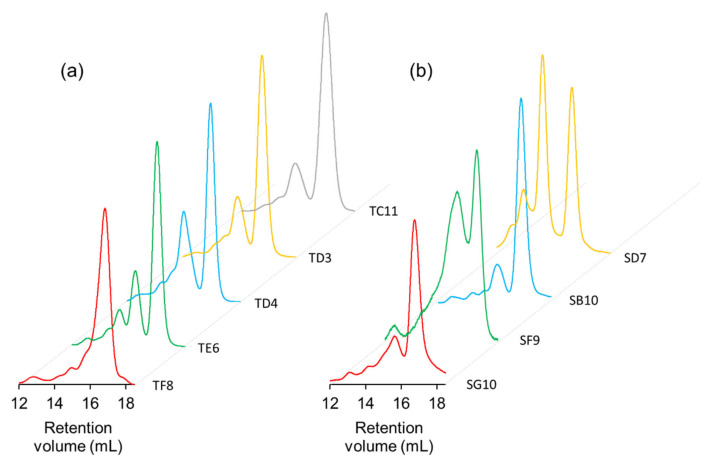
SEC analysis of scFv clones from (**a**) OPAL-T and (**b**) OPAL-S. Monomeric scFv was eluted at retention volume ~17 mL. Clones show varying degrees of dimerization/aggregation (approximately 14–16 mL retention volume).

**Table 1 ijms-23-06255-t001:** Sizes estimated by transformation titers and dot blot-positive percentages of OPAL-T sub-libraries.

CDR-H3 Length (aa)	Kappa Light Chain		Lambda Light Chain	
	Size	Dot Blot%	Size	Dot Blot%
9	3.7 × 10^8^	31.8	0.9 × 10^9^	54.5
10	6.2 × 10^8^	54.5	1.5 × 10^9^	45.5
11	4.8 × 10^8^	54.5	1.4 × 10^9^	45.5
12	5.0 × 10^8^	31.8	1.4 × 10^9^	50.0
13	2.2 × 10^8^	18.2	1.1 × 10^9^	50.0
14	4.1 × 10^8^	54.5	1.0 × 10^9^	59.1
15	3.5 × 10^8^	45.5	1.4 × 10^9^	50.0
16	5.1 × 10^8^	36.4	1.9 × 10^9^	54.5
Total	3.4 × 10^9^	40.9 (72/176)	1.1 × 10^10^	51.1 (90/176)

**Table 2 ijms-23-06255-t002:** Comparison of the panning performances between OPAL-S and OPAL-T using a panel of antigens.

Antigen	Library	ELISA Positive/Screened		Unique/Sequenced	
		Round 3	Round 4	Round 3	Round 4
hNinj1 ^1^	OPAL-T-κ	91/94	93/94	2/5	ND ^9^
	OPAL-T-λ	52/94	94/94	4/6	2/6
	OPAL-S-κ	66/94	ND ^9^	1/4	ND ^9^
	OPAL-S-λ	4/94	8/94	3/7	2/3
CARS ^2^	OPAL-T-κ	91/94	91/94	3/8	2/10
	OPAL-T-λ	91/94	92/94	7/8	8/8
	OPAL-S-κ	46/94	89/94	10/12	5/9
	OPAL-S-λ	91/94	94/94	5/10	3/8
AgX ^3^	OPAL-T-κ	48/94	24/94	4/8	6/9
	OPAL-T-λ	19/94	10/94	10/13	7/7
	OPAL-S-κ	16/94	2/94	11/13	2/2
	OPAL-S-λ	5/94	4/94	3/3	1/1
BCMA ^4^	OPAL-T-κ		ND ^9^		ND ^9^
	OPAL-T-λ		90/94		5/11
	OPAL-S-κ		2/47		2/2
	OPAL-S-λ		1/47		1/1
CD22 ^5^	OPAL-T-κ		ND ^9^		ND ^9^
	OPAL-T-λ		92/94		3/8
	OPAL-S-κ		12/94		2/6
	OPAL-S-λ		27/94		2/8
HEWL ^6^	OPAL-T-κ		1/188		1/1
	OPAL-T-λ		4/188		3/4
	OPAL-S-κ		24/94		1/6
	OPAL-S-λ		45/94		5/25
SARS ^7^	OPAL-T-λ		80/94		4/5
	OPAL-S-κ+λ		16/94		3/14
AIMP1 ^8^	OPAL-T-λ		86/94		5/5
	OPAL-S-κ+λ		66/94		5/8

^1^ hNinj1, human ninjurin-1; ^2^ CARS, cysteinyl tRNA synthetase; ^3^ AgX, undisclosed antigen; ^4^ BCMA, B cell maturation antigen; ^5^ CD22, cluster of differentiation 22; ^6^ HEWL, hen egg white lysozyme; ^7^ SARS, seryl tRNA synthetase; ^8^ AIMP1, aminoacyl tRNA synthetase interacting multifunctional protein 1; ^9^ ND, not determined.

**Table 3 ijms-23-06255-t003:** Binding kinetics parameters of target-binding scFvs selected from OPAL-T.

Antigen-Clone	*k*_on_ (M^−1^s^−1^)	*k*_off_ (s^−1^)	*K*_D_ (M)
CARS-B6	1.0 × 10^5^	4.2 × 10^−3^	4.1 × 10^−8^
CARS-D7	1.6 × 10^4^	2.2 × 10^−3^	1.3 × 10^−7^
CARS-D11	2.8 × 10^4^	2.3 × 10^−3^	8.2 × 10^−8^
CARS-F4	3.6 × 10^4^	3.7 × 10^−3^	1.1 × 10^−7^
BCMA-A3	8.7 × 10^4^	2.6 × 10^−3^	3.0 × 10^−8^
BCMA-B5	7.5 × 10^4^	2.7 × 10^−3^	3.7 × 10^−8^
BCMA-D11	2.4 × 10^4^	1.9 × 10^−3^	7.6 × 10^−8^
AIMP1-C6	5.4 × 10^4^	1.5 × 10^−3^	2.8 × 10^−8^
AIMP1-D4	3.2 × 10^4^	2.5 × 10^−3^	7.6 × 10^−8^
AIMP1-E7	1.8 ×10^4^	2.0 × 10^−3^	1.1 × 10^−7^
SARS-D6	5.8 × 10^4^	1.1 × 10^−3^	2.0 × 10^−8^
SARS-F4	1.5 × 10^5^	4.0 × 10^−3^	2.7 × 10^−8^
CD22-D1	1.4 ×10^4^	6.2 × 10^−4^	4.5 × 10^−8^
HEWL-H5	5.0 ×10^3^	1.8 × 10^−3^	3.6 × 10^−7^

**Table 4 ijms-23-06255-t004:** Purified yields and monomeric percentages of anti-hNinj1 scFv clones from OPAL-T and OPAL-S.

Library	Clone	Yield (mg/L Culture)	% Monomer
OPAL-T	TF8	21	86
	TE6	32	65
	TD4	22	58
	TD3	32	70
	TC11	26	77
	OPAL-T average ± S.D.	26.6 ± 5.3	71.2 ± 10.1
OPAL-S	SG10’	12.4	75
	SF9	4.1	34
	SB10	5.8	79
	SD7	23.3	35
	OPAL-S average ± S.D.	11.4 ± 8.7	55.8 ± 24.6

## Data Availability

Not Applicable.

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
