# Peer review of "A Novel Synthetic Antibody Library with Complementarity-Determining Region Diversities Designed for an Improved Amplification Profile"

_ijms, 2022, doi:10.3390/ijms23116255_

Round 1

Reviewer 1 Report

Bai and colleagues present the construction of an improved version of their synthetic antibody phage display library OPAL-S. The work describes the process of optimization of the previous library and is in particular focused towards the reduction of the often overlooked bias introduced upon the amplification phase during panning. The use of a well-know Protein A-bound VH3 sequences allows the author to select properly folded scFv and implement the sequence characteristics of these clones in the new library OPAL-T. An analysis of the generated library follows, as well as an effort in comparing panning outputs against different targets with both libraries.

The paper and the strategy present are indeed interesting and the new library obtained is wider (two order of magnitude more than the previous one) and appear to have more expressing clones. However, for the benefit of the readership there are some points to be addressed.

Major points:

  • The theory behind the use of a Protein A panning for both scFv selection and improved displaying scFv is clear for people in the field. The manuscript would greatly improve if this is clearly outlined in the introduction. Also has this technque already been used?
  • Results presented in Fig. 2 are described in a rapid way, only pointing to the biggest differences seen for CDR-H2. However other notable differences between the pre- and post-selection percentage are present in almost all CDRs, except CDR-lambdaL2. The families that are enriched or depleted after selection should be commented: some points (i.e. gene families) show a shift of 10% or more, which cannot be neglected.
  • In Fig. 3 the numbering scheme adopted for the CDR-H3 region should be mentioned. As well, a comparison with the natural amino acid frequency at each position should be presented to draw better conclusion on the quality of the new library: did the protein-A panning forced a deviation from the natural occurrence of amino acids in CDR-H3?
  • Do the author have an explanation why in the combined final OPAL-T library there is a much reduced percentage dot blot-positive clones compared to the single CDR libraries? The difference is quite remarkable, since single CDR libraries seem to be very well behaved, but once pooled there is quite a loss.
  • In the section of the proofreading panning or in the discussion it has to be mentioned that the theoretical size of OPAL-T is two order of magnitude higher than that of OPAL-S, which can also explain the better panning outputs.
  • About Table 3, especially for the antigens where 3 and 4 panning rounds have been made: can the author clarify if the unique final scFv sequences are different between the 3 and 4 panning round selection campaigns or are the same?
  • Materials and Methods section: please add information on Protein A used for the panning of the library, how it was produced and if was the wild type Protein A of S. aureus.
  • There is no description on how scFv have been produced, purified and also no description of SPR analysis instrument and parameters
  • Can the author clarify if the use of an codon-optimized synthesized VH3-23 scaffold affected the library expression? Was the same optimization carried out for OPAL-S library?
  • All throughout the text, the bacterial species names are not in italics.

Minor comments:

  • A tone down of the sentence “the highly enriched clones often are rapidly amplified sequences with low or no binding activity” is suggested, otherwise it seems that antibody phage display is highly unsuccessful.
  • Line 40: should read techonological platform
  • Indicate purity of the oligos used (line 406)
  • Codon optimized VH3-23 has been used. Optimized for which organism?

Reviewer 2 Report

The manuscript by Bai et al. describes the design and construction of a synthetic scFv phage display library (OPAL-T) with CDR diversities aimed at improved amplification efficiency. The authors had previously constructed a synthetic scFv library (OPAL-S) which served as the starting point in the current study to construct the improved OPAL-T. To this end, the authors panned OPAL-S against protein A for three rounds and analyzed the library repertoires before and after the panning by NGS. The enrichment or depletion patterns of CDR sequences after panning served as the basis for the design of OPAL-T. OPAL-T and OPAL-S were panned in parallel against a panel of antigens. Better performance of the second generation library (OPAL-T) was concluded based on the higher number of ELISA+ clones and unique “binders”. The authors then selected scFvs identified from the OPAL-T library and determined their kinetic and equilibrium binding constants by SPR. The manuscript in its current form is not accepted for publication, mainly because key experiments (see A2 and A3) which would provide conclusive and convincing evidence for the better performance of OPAL-T over OVAL-S is missing. Upon providing satisfactory data as requested, the manuscript will be publication worthy.

A. Conclusive and convincing evidence showing OPAL-T performed better than OPAL-S is missing:

  1. # of unique sequences is more or less the same for both OPAL-T and OPAL-S libraries.
  2. Importantly, the authors only selected OPAL-T clones and determined their affinities by SPR. They should have also selected a similar number of scFvs from the OPAL-S library and determined their affinities and validate their binding by SPR. Here, the yields and affinities of the binders characterized by SPR could be used as criteria to rank the two libraries. Therefore, authors need to perform the aforementioned SPR experiments. They should also show all sensorgrams in the Supp. Data and representatives in the main text.
  3. Aside from affinity, expression and solubility/aggregation resistance of positive clones identified by panning are also important criteria by which the two libraries should have been compared. The authors should assess the expression levels as well as solubility and aggregation resistance (determined by analytical size-exclusion chromatography [SEC]) of similar number of both OPAL-T and OPAL-S scFvs. Expression yield and SEC data (as tables and figures) should be included in the main text.

B. Methods

  1. While clones were expressed and purified for SPR binding analysis, the relevant methods are missing. Include methods for the expression and purification of scFvs.
  2. Clones are assessed by SPR but relevant methods are missing. Include SPR method.
  3. Dot blot results in the Supplementary Data lacks method descriptions. Include dot blot method.

C. Other comments:

  1. add “OPAL-T” after “Library construction” in Figure 1.
  2. Figure 2: replace “(B)” with “(b)”.
  3. Figure 3c: specify if H95, H96 and so on are based on sequential numbering or other numbering systems, e.g., Kabat.
  4. Section 4.3: define “ES” in the opening sentence. In the formula below it, remove “Enrichment score”.

Reviewer 3 Report

Authors constructed and validated scFv library with CDR diversities for improved amplification efficiency. The topic of this manuscript is well fit to the International Journal of Molecular Sciences and the logic of manuscript is scientifically sound. I would recommend to publish this article after minor modification as below:

  1. Abstract contains lots of unnecessary introduction. Line 12~18 need to be shortened.
  2. Table 1 and 2 can be combined into one table.

Author Response

Dear Reviewer:

Thank you for kindly reviewing the manuscript. Corrections has been made according to the reviewer’s comments as summarized below (blue).

Authors constructed and validated scFv library with CDR diversities for improved amplification efficiency. The topic of this manuscript is well fit to the International Journal of Molecular Sciences and the logic of manuscript is scientifically sound. I would recommend to publish this article after minor modification as below:

  1. Abstract contains lots of unnecessary introduction. Line 12~18 need to be shortened.

Thank you for the comment. Lines 12-18 in the original manuscript has been shortened.

  1. Table 1 and 2 can be combined into one table.

Thank you for the comment. Correction has been made as requested.

Round 2

Reviewer 2 Report

None